# Situating Wikipedia as a health information resource in various contexts: A scoping review

Denise A. Smith [1,2] *

**1** Health Sciences Library, McMaster University, Hamilton, Ontario, Canada, **2** Faculty of Information & Media Studies, Western University, London, Ontario, Canada

* dsmit94@uwo.ca

**Data Availability Statement:** All relevant data are within the manuscript in Appendices B through E, inclusive.

**Funding:** The Wikimedia Foundation (awarded to DS) https://meta.wikimedia.org/wiki/Grants: Project/Rapid/Mcbrarian/Situating_Wikipedia_as_ a_health_information_resource_in_various_

## Abstract

### Background

Wikipedia's health content is the most frequently visited resource for health information on the internet. While the literature provides strong evidence for its high usage, a comprehensive literature review of Wikipedia's role within the health context has not yet been reported.

### Objective

To conduct a comprehensive review of peer-reviewed, published literature to learn what the existing body of literature says about Wikipedia as a health information resource and what publication trends exist, if any.

### Methods

A comprehensive literature search in OVID Medline, OVID Embase, CINAHL, LISTA, Wilson's Web, AMED, and Web of Science was performed. Through a two-stage screening process, records were excluded if: Wikipedia was not a major or exclusive focus of the article; Wikipedia was not discussed within the context of a health or medical topic; the article was not available in English, the article was irretrievable, or; the article was a letter, commentary, editorial, or popular media article.

### Results

89 articles and conference proceedings were selected for inclusion in the review. Four categories of literature emerged: 1) studies that situate Wikipedia as a health information resource; 2) investigations into the quality of Wikipedia, 3) explorations of the utility of Wikipedia in education, and 4) studies that demonstrate the utility of Wikipedia in research.

### Conclusion

The literature positions Wikipedia as a prominent health information resource in various contexts for the public, patients, students, and practitioners seeking health information online. Wikipedia's health content is accessed frequently, and its pages regularly rank highly in Google search results. While Wikipedia itself is well into its second decade, the academic discourse around Wikipedia within the context of health is still young and the academic

contexts:_A_systematic_review_of_the_literature
This grant was awarded to DS after the work was complete and the manuscript had been submitted. The funders had no role in study design, data collection and analysis, decision to publish, or preparation of the manuscript.

**Competing interests:** I have read the journal's policy and the author of this manuscript has the following competing interest: The author is a Wikipedia contributor and advocates for Wikipedia authorship to health workers, students, and faculty at McMaster University.

**Abbreviations:** JMIR, Journal of Medical Internet Research; SR, Systematic review(s); OA, Open Access.

literature is limited when attempts are made to understand Wikipedia as a health information resource. Possibilities for future research will be discussed.

## Introduction

"Imagine a world in which every single person has free access to the sum of all medical knowledge" [1]. This is what WikiProject Medicine, a community of editors who develop the health care content in Wikipedia, is doing and it hasn't gone unnoticed. Wikipedia is the most frequently accessed resource for health information: its English language medical content has received more unique pageviews than any other health information resource online [2]. Of course, any discussion of Wikipedia would be incomplete without acknowledgement of the stigma attached to it. It is common knowledge that there is a general wariness around using Wikipedia [3]. However, a paradigm-shifting study conducted by *Nature* found that Wikipedia and *Encyclopedia Britannica* were comparable in the number of errors found in a sample of entries from each resource [4]. This finding has not been challenged since its publication and was vital in the ongoing effort to de-stigmatize Wikipedia.

Further, an exploration of what might motivate editors to contribute to health-related articles on Wikipedia found that editing Wikipedia is influenced predominantly by an editor's inherent values–a belief that all knowledge should be free; and a sense of obligation that they have a duty to fix Wikipedia as health workers [5]. This single study offers unique insight into the motivations of those who voluntarily edit existing content, translate studies and reviews into plain language, or translate content to languages other than English. While the findings of this study are not generalizable, they imply that behind the pages of Wikipedia is a community of (mostly) altruistic editors motivated by a shared core value that information should be available and accessible to all communities.

Today Wikipedia is the 7th most frequently accessed web site in the United States and 8th in Canada [6,7]. For perspective, English Wikipedia alone averages approximately 830 million unique visits per month globally [8]. Its content is regularly one of the top results in Google searches, with about 93% of clicks on Wikipedia directed from Google [9]. However, efforts to understand, in depth, Wikipedia's role in the landscape of health information resources are limited. While systematic reviews of the scholarly literature covering Wikipedia have been conducted [10,11] a comprehensive literature review that focuses specifically on the treatment of Wikipedia in the academic health literature has not yet been reported. This scoping reviews asks:

1. What does the existing body of literature say about Wikipedia as a health information resource? and,

2. What publication trends exist, if any, among health-related articles about Wikipedia?

## Methods

### Search strategy

A scoping review is a comprehensive and robust literature review that diverges from the systematic review in its aim to "summarize and disseminate [existing] research findings", "identify research gaps in the existing literature", and may not incorporate strict criteria, such as study design, as a requirement for inclusion in the review [12]. The literature search and

retrieval process for a scoping review can be iterative in its execution and relies heavily on mapping concepts or themes within the body of literature. As a result, scoping reviews may place an emphasis on producing a highly sensitive search rather than a search with more balance between specificity and sensitivity, as is done in systematic literature reviews. To maximize the sensitivity of her literature search, the author developed and executed a search strategy (S1 Appendix) in June 2019 to retrieve any article where the term "Wikipedia" appears in either the title or the abstract of any article indexed in OVID Medline, OVID Embase, EBSCO CINAHL, and OVID AMED. As these three databases are clinically focused, it is likely that the term "Wikipedia" will be retrieved within a health context. The author also searched for the term "Wikipedia" in the titles or abstracts of all articles indexed in EBSCO Library and Information Science and Technology Abstracts (LISTA) and EBSCO Library Literature and Information Science Full Text (Wilson Web).

A literature search in Web of Science presented a special conundrum due to the limitations of its advanced search interface where it is not possible to perform keyword searches for terms appearing in the abstract. As a result, the author chose to conduct a sweeping search for the term "Wikipedia" in any Web of Science record, which she followed up with a manual screening of all results to remove irrelevant content. Content was considered irrelevant if the term Wikipedia did not appear in the title or abstract or if the title or abstract indicated the article was not health related.

The author's exclusion of search terms related to health and medicine was intentional. While one may be inclined to use health-related terms in the search strategy, the breadth of terms that would need to be included in order to capture all health-related concepts would be prohibitively time-consuming and prone to human error. In order to get a sense of how and in what health fields Wikipedia is discussed in the literature, the author decided an attempt to create an exhaustive search string to retrieve anything health-related posed too great a risk. Instead, the author elected to conduct a broad search and use exclusion criteria to manually eliminate irrelevant records not related to health and medicine. A protocol for this review was not published or registered.

## Study selection

All records were exported to a free citation manager (Zotero) and imported into systematic review (SR) software (DistillerSR). Using the SR software, database records were screened first by title and abstract. A second round of review screened the full text of all records. Records were excluded if English Wikipedia was not a major or exclusive focus of the article; English Wikipedia was not discussed within the context of a health or medical topic; the article was not available in English (no funds were available to support translation for this review); or the article was a letter, commentary, editorial, or popular media article.

In some cases, a conference proceeding record and a published article record were retrieved for the same study. In these instances, the conference proceeding was excluded. Some articles that met the criteria requiring a discussion of Wikipedia's health or medical content were excluded because Wikipedia was not a major focus of the paper and the treatment of Wikipedia's health content was insufficient to merit inclusion.

8,030 records were retrieved in the literature search. After manually removing irrelevant studies from the Web of Science search results (n = 4,400) and removing duplicates (n = 1,417), the literature search retrieved 2,213 unique results (Fig 1) that were subsequently screened for relevance based on title and abstract content. 1,815 articles were excluded in the first round of screening. After a review of the full-text using the exclusion criteria outlined

Flow Diagram

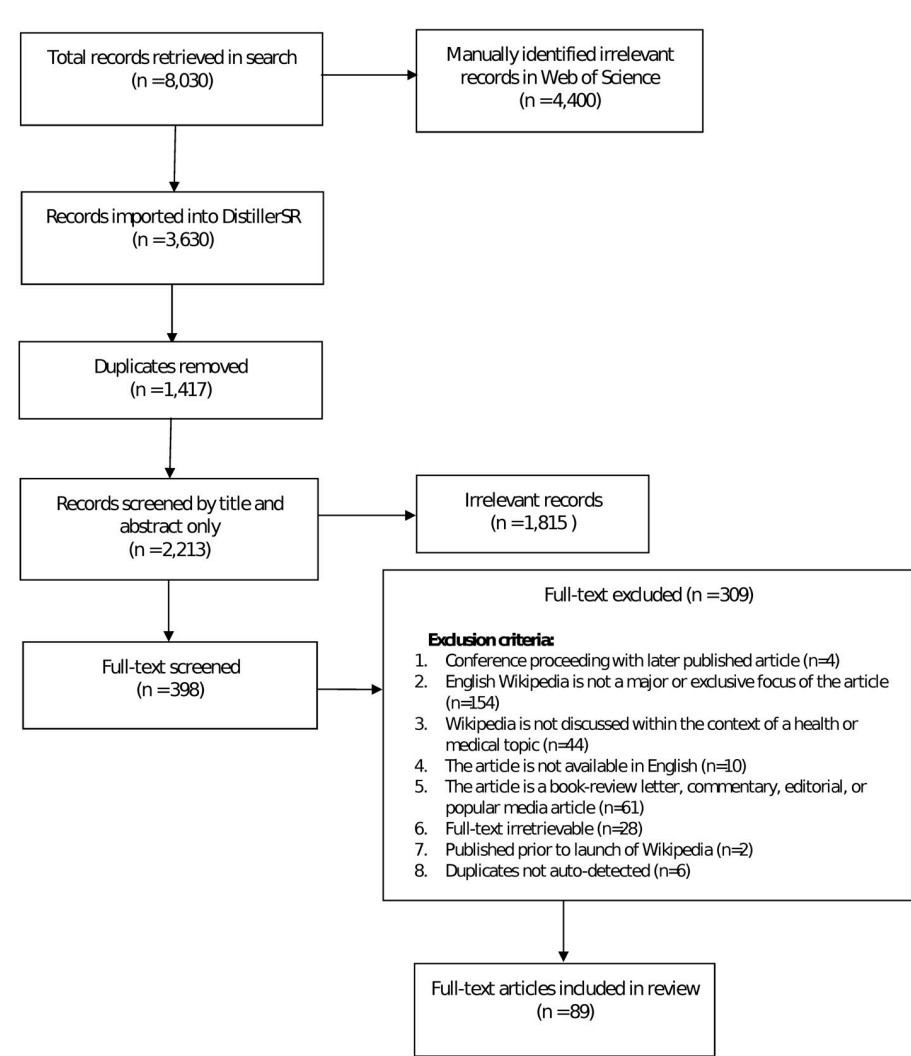

Based on: Moher D, Liberati A, Tetzlaff J, Altman DG, The PRISMA Group (2009). *Preferred Reporting Items for Systematic Reviews and Meta-Analyses: The PRISMA Statement. PLoS Med 6(7): e1000097. doi:10.1371/journal.pmed1000097*

**Fig 1. PRISMA flow diagram of studies selected for inclusion.**

above an additional 309 articles were excluded. As a result, 89 articles and conference proceedings were selected for inclusion in this scoping review.

## Thematic analysis

The objective of the author's thematic analysis was to inductively analyse and identify any patterns, themes, or trends, that emerged through a reading of the full-text of all 89 studies included in the review. The author chose an inductive approach to remain open to any themes that may emerge from the literature as she read.

The first step in her thematic analysis included the generation of an annotated bibliography. The exercise of engaging with each article and annotating its methods, findings, and research questions, allowed patterns to naturally emerge. The thematic analysis was an organic process that unfolded as the author read each paper. As patterns emerged, the author maintained a

table of themes and patterns. Upon completion of the annotated bibliography, the author organized each emergent theme into one of four larger categories in which the articles were placed:

1. Articles with a general focus on situating Wikipedia as a health information resource (S2 Appendix);

2. Assessments of Wikipedia's quality (S3 Appendix);

3. Wikipedia's utility for education (S4 Appendix), and;

4. Wikipedia's utility for research (S5 Appendix).

## Results

### Situating Wikipedia as a health information resource

Approximately 1% (n = 9) of the 89 included articles focus on generally situating Wikipedia as a health information resource (Fig 2) by measuring its readership in comparison to other online health information sources that are already validated by the health or information community. Despite the small relative size of this category of the literature, these studies (S2 Appendix, Table 1) position Wikipedia as a heavily used health information resource. They do not focus on any particular context or user population.

Publications that position Wikipedia as a health information resource demonstrate this by referencing its frequency of use. Some leverage this data to call on the medical community to contribute to Wikipedia's health and medical content. The Pew Research Centre collected evidence that establishes the high-use of Wikipedia for health information among the public, physicians, and medical students [13] and Heilman, et al. question the relevance of traditional peer-reviewed articles when Wikipedia's health content can be updated in real-time, has a global audience, and is written in plain, accessible language [13].

A more recent investigation finds that Wikipedia's medical content is now consistently superior to its non-medical content with 83% of its articles rated as B-Class, the highest quality rating an article can receive without undergoing internal peer-review [14]. Overall, the literature within this theme attempts to destigmatize Wikipedia by addressing its limitations as a health information resource and whether solutions for these challenges are a possibility [15].

Situating Wikipedia as a health information resource is also verified by its relative ubiquity in online searches. Although it is based on data from 2008, a 2011 study finds that Wikipedia is the first result for most online generic drug name searches [16]. Further, its general capacity is also helpful in positioning Wikipedia as a health information resource. At the end of 2015, Wikipedia had 155,085 medical articles, approximately 18% of which were in English Wikipedia. This content is supported by more than 950,000 references and the most commonly cited resources are top-tier medical journals such as, New England Journal of Medicine, The Lancet, Nature, and Cochrane Systematic Reviews [15]. A single study places Wikipedia as one of the most heavily accessed sources of online health information based on aggregated pageview data of Wikipedia articles under the umbrella of WikiProject Medicine [2]. It finds Wikipedia to be equal to that of the National Institutes of Health web site and surpasses WebMD, another dominant health information web site.

Other investigations into the validity of Wikipedia as a health information source include explorations of the opportunity Wikipedia's health content offers globally [13]. For example, during the 2014 Ebola outbreak in West Africa, Wikipedia's Ebola content was drastically updated, translated to more than 100 different languages, and was viewed more than 89 million times that year [14]. However, the most influential and widely cited study in the literature

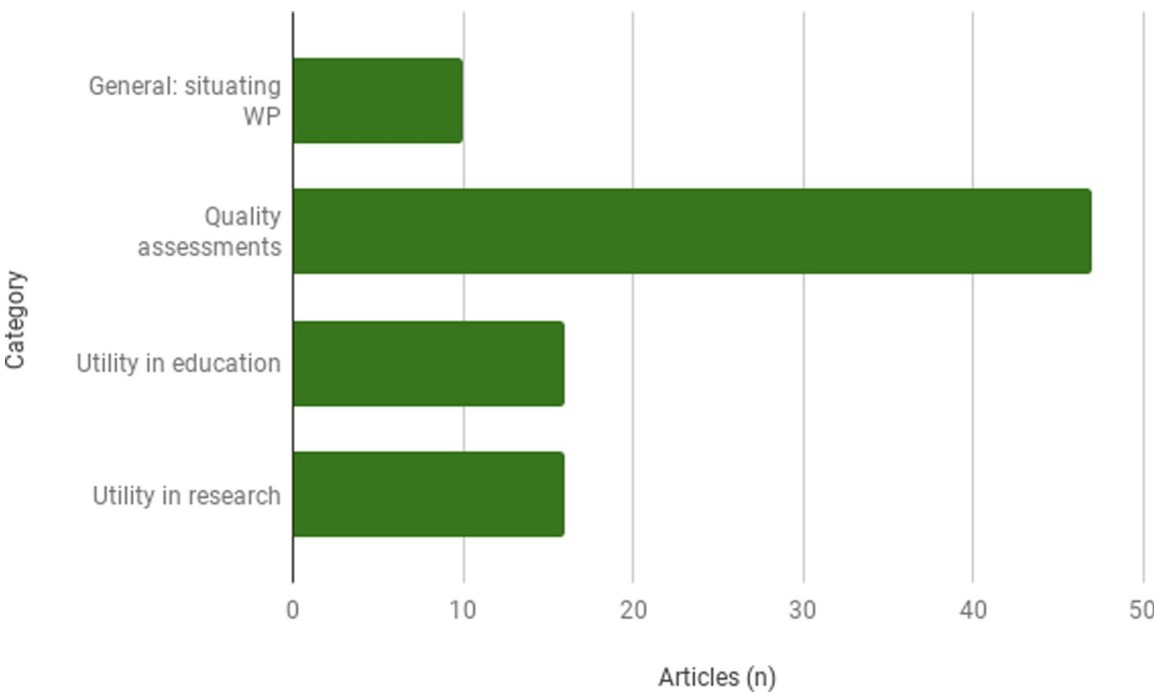

**Fig 2. Distribution of relevant studies by category.**

places Wikipedia at the forefront of online health information resources [17]. Laurent and Vickers (2009) find a statistically significant difference, in which Wikipedia takes prominence, between the pageviews of Medline Plus and Wikipedia for the same topics [17]. This seminal study positions Wikipedia as prominent among the existing complement of online health information resources.

## Quality of Wikipedia's health content

Assessments of Wikipedia's quality are categorized by the population of concern to the study. That is, whether the quality assessment is conducted with a general information consumer or patient in mind, a health professional, a student, or whether a user of Wikipedia is not clearly identified in the study. The focus on assessments of the quality of Wikipedia's health content within the literature is overwhelming. Despite efforts to affirm Wikipedia's value as a source for health information, questions about its quality dominate the conversation. 53% (n = 47) of the articles included in this study assess the quality of Wikipedia's health content (Fig 2). In this review, studies of Wikipedia's quality include assessments of readability and reliability, wherein reliability may include any of the following measures: accuracy, comprehensiveness, completeness of coverage, number or quality of references, and currency.

**For consumers.** It is well established that the public uses the internet, and by extension Wikipedia, to find and consume health information. Of the articles included in this review, 19% (n = 17) were concerned with the suitability of Wikipedia's health content for patients or consumers. These studies (S3 Appendix, Table 2) assess readability, reliability, and accuracy or completeness, but specifically discuss their findings in relation to the public consumer or patient. Of those that include results–some conference proceedings do not–they generally agree that Wikipedia is suitable for patients [18–20]. Further, a 2010 study claims that while Wikipedia is not the superior resource, its findings suggest that it is the preferred resource

[21]. Therefore, it could be suggested that enquiries into Wikipedia's suitability for patients may have limited impact if patients are likely to prefer Wikipedia anyhow. Unfortunately, these assessments of suitability for patients are also unlikely to be relevant given that their publication dates range between 2008 and 2013.

Some unique studies that examine the quality of Wikipedia include Skancke and Svendsen's (2017) investigation into who edits Wikipedia's content on medicines. Skancke and Svendsen explore the impact and potential implications associated with pharmaceutical representatives editing Wikipedia [22]. Another study investigates the impact of the contributor's experience on the quality of Wikipedia articles through an evaluation of edits [23] and Cozza, Petrocchi and Spognardi use natural language processing software to automate the assessment of Wikipedia articles [24]. However, most studies focus on the information in Wikipedia about a specific health speciality. These include gastroenterology [18], nephrology [19], autoimmune disorders [25], medicinal drugs or herbal supplements [26–31], pathology informatics [32], surgery [33,34], toxicology [35,36], nutrition [37–39], complementary and alternative medicine [40], cancer [20,21,41,42], hearing loss [43], and mental health or the brain [44,45].

Two distinctive studies describe leveraging Wikipedia's content in order to improve patient care. Martin-Carreras & Kahn (2019) describe an online patient-oriented information resource in radiology that is now integrated with Wikipedia to pull radiology images from the Wikimedia Commons into their database [46]. Another study describes how it used Wikipedia to mine consumer health vocabulary by finding synonyms for medical terminology in order to improve communication between practitioners and patients [47]. While the content is not necessarily evaluated for consumption, both of these studies are examples of how Wikipedia's health content could be used in creative ways to complement existing resources maintained by experts and improve access to information or patient care.

**For students.**   Less than 1% (n = 8) focused on the suitability of quality of Wikipedia for student use (S3 Appendix, Table 3). Herbert, et al (2015) present evidence that suggests most medical students use Wikipedia at a moderate or high rate (67%), but this investigation reports a response rate of 21% so the findings cannot be generalized [48]. Interestingly this study also finds that of the participants, 65% did not know how to revise Wikipedia when they encountered an error. This begs the question of whether faculty should follow the lead of Queen's University [49] and USCF [50] and instruct medical students to edit Wikipedia so that they can revise errors when they inevitably use the resource. In a more methodologically sound study, Judd and Kennedy (2011) find that medical students used Google in 69% of biomedical sessions in a computer lab and Wikipedia in 51% of those same sessions [51]. While the study also notes an interesting trajectory in which students' reliance on Wikipedia decreases each year from first year to third year, actual Wikipedia usage remains prominent throughout students' progression through the curriculum.

Assessments of Wikipedia's suitability as a learning tool [52–59] range in topic and methods and again, encounter similar challenges to the quality assessment studies discussed earlier. Information sources used as a point of comparison and methodologies are not necessarily appropriate, Wikipedia's dynamic nature reduces the reliability of the studies, and the findings vary from one study to another. Some conclude Wikipedia is suitable for medical students [53,57], but most conclude it is not [53–56,58,59].

**For professionals.**   A small set of studies (n = 5) framed evaluations of Wikipedia's health content within the context of its suitability for use by health care workers (S3 Appendix, Table 4). Park, Masupe, Joseph, et al (2016) report that Botswanan health care workers' perceptions of Wikipedia's quality is divisive at best. In this study, Wikipedia's value was based more on its consistent availability, through the now defunct Wikipedia Zero project [60], than the quality of its medical content [61].

Within specific medical contexts, the findings are inconsistent. For example, as a surgical reference Wikipedia is found to be accurate and appropriate resource, albeit incomplete [33]. Conversely, the drug information on Wikipedia is deemed variable in comparison to Micromedex and therefore considered an inappropriate drug reference for professionals [31]. Within the professional context, the literature also discusses efforts to improve the quality of Wikipedia. For example, Morata and Lum (2018) describe a partnership between the U.S. National Institute for Occupational Safety and Health (NIOSH) and Wikipedia with the goal to improve Wikipedia's occupational safety and health content [62]. NIOSH developed and continues to manage WikiProject Occupational Safety and Health and NIOSH also participations in a program that encourages students to contribute to Wikipedia. This partnership could be interpreted as recognition of Wikipedia's influence and broad audience.

Wikipedia's prevalence in scholarly communication has also caught the attention of researchers. In a unique departure from assessing Wikipedia's citations, Bould, et al. (2014) examine the prevalence of Wikipedia citations in peer-reviewed health sciences journals indexed in PubMed, Medline, or Embase [63]. The study finds a small but growing frequency of Wikipedia citations in peer-reviewed health sciences literature and that most of these citations occur after 2010. The majority of these citations are used for descriptions or definitions. This growth in referencing Wikipedia in peer-reviewed literature could be an indication of the growing reliability of Wikipedia's health content.

**General assessments of quality.** General assessments of the quality of Wikipedia's health information, those that do not take into consideration the impact of its quality on a specific population, comprise approximately 18% (n = 16) of the articles included in this review (S3 Appendix, Table 5). Despite claims that English Wikipedia's health content is written in plain, accessible language, assessments of its quality, regardless of context, share a common finding: while Wikipedia does well to remain current [26,41,44], its medical content uses technical terms that result in readability levels too low to accommodate the very people it stands to benefit most. The readability of the easiest articles has been reported to be around ninth grade [25,30,40]. Some studies find that Wikipedia's medical content requires at least a college reading level or is considered highly difficult [34,43]. Other studies simply report Wikipedia's readability as higher than that of other health information sources such as WebMD, MedlinePlus, or MayoClinic [44,64].

## Utility in education

Wikipedia's utility in education is also considered within the context of different populations, ranging from its general use in educational settings to its deliberate inclusion in course or program curricula, specifically as an educational tool for evidence-based practice or information literacy, or both. The term "education" here applies to students enrolled in a formal medical or health education program, professionals enrolled in a continuing education program, workshop, or course, and professionals who use Wikipedia to inform their everyday practice. Approximately 18% (n = 16) of the literature in this review addresses the utility of Wikipedia in health education within the above three contexts.

Explorations into the various roles of Wikipedia in health and medical education have a strong presence within the literature (S4 Appendix, Table 6), which includes editing Wikipedia as an educational tool in the classroom [49,50,52,65–68], measuring student interactions with Wikipedia [48,51,53,69], and its utility in knowledge sharing between professionals or for daily reference in everyday practice [52,70–73].

The first documented for-credit course at a medical school for editing Wikipedia was offered by University of California, San Francisco in 2013 [50]. The first reported use of

Wikipedia in a medical course, as a single assignment, is from 2011 and it required students to edit neuroscience stub articles [66]. Since then, the inclusion of editing Wikipedia in health education courses has crossed a number of health disciplines. Pharmacy students have been asked to write content in Wikipedia as an alternative to composing pharmaceutical drug monographs [65] and a gerontology instructor identified gaps in Wikipedia's content on aging and assigned students the task of using their term papers to edit it [67].

A more recent example comes from Queen's University where editing Wikipedia is integrated into the first-year evidence-based medicine curriculum [49]. It is a strong example of the utility of Wikipedia to enforce critical skills in medicine, where students are instructed in evidence-based medicine and are subsequently asked to apply those skills in a project that has real-world impact with relatively low risk. Finally, in a survey of first year students from the same program [69], the authors find that students recognize the utility of Wikipedia after editing it and developed a greater appreciation for it after completing their evidence-based medicine projects. Interestingly, the majority of survey participants also indicated that they would be likely to use Wikipedia during medical school and their subsequent residencies but would not necessarily recommend it to patients.

One distinguishing study is a randomized control trial from 2017 that compares Wikipedia to UpToDate and a digital textbook. It tests for short-term knowledge acquisition among a sample of first and second-year medical students from Canadian medical schools [74]. This two-year parallel arm RCT finds a statistically significant difference in knowledge acquisition between the textbook group and the Wikipedia group and an observed trend of better performance from the Wikipedia group compared to the UpToDate group, albeit the difference between the two groups is not found to be statistically significant. This study is unique because it employs traditional scientific rigour, a rarity here. The authors hypothesize that Wikipedia's content may impose a smaller cognitive load than the digital textbook, which could have been a contributor to a significantly better performance on post-tests in the Wikipedia group than in the digital textbook group. It presents a strong case against the traditional discouragement of Wikipedia use in medical education and instead presents evidence to support its potential utility in the education of medical students.

With respect to adult education, continuing education, or professional practice (S4 Appendix, Table 7), Lieberthal and Leon (2015) report on having adult health economics students post their written assignments to Wikipedia [52]. Additionally, genomics experts use and contribute to the GeneWiki, a collection of Wikipedia entries where each entry is related to a specific human gene, which aims to comprehensively annotate gene functions for each known human gene [70]. The GeneWiki has been integrated into the larger body of Wikipedia articles [70] and is presented as a solution to the previous problem biomedical researchers faced in producing review articles for each of the 20,000 human genes [71]. Further, the GeneWiki itself has been mined for its gene annotations in an effort to generate a gene ontology and a disease ontology [72]. Finally, a study from New Zealand, while citing a low participation rate, suggests that Wikipedia use by interns, residents, and consultants is common to aid with memory recall or to provide a fast overview [73].

## Utility in research

Wikipedia's role in health or medical research is also discussed within the literature within two different contexts: in human information practices research, as a data source for bibliometric and epidemiological research, and in epidemiology research. It's utility in epidemiological research constitutes half (n = 8) of the 18% (n = 16) of articles that focus on Wikipedia's utility in health-related research.

**Bibliometric and information-seeking behaviour studies.** As with assessments of quality and suitability for medical students or patients, the small contingent of infodemiology research (S5 Appendix, Table 8) is also heavily reliant upon subject specific Wikipedia content. These include movement disorders [75], neurological disorders [76,77], hemophilia [78], and critical care [73]. While this is not a robust field of literature some findings are worth noting. Namely, that pageviews in Wikipedia spiked after a celebrity's announcement of diagnosis, which suggests Wikipedia might play an important role in the public consumption of health information [77]. A single study of Wikipedia's dominance in search engine results suggests that while specialist web sites may have more depth, they may be at higher risk than Wikipedia of being missed in more general searches due to Wikipedia's prominence in search engine retrievals [78].

Bibliometric studies (S5 Appendix, Table 9) have been used to assess whether Wikipedia is cited in academic literature and its influence on the popularity of, or access to, high-impact journals, or to measure article impact. Wikipedia is positioned as a key alternative resource for the measurement of journal or article impact [79], to identify influential medical journals [80], or to act as a "gateway" to traditionally published health research, given that approximately 69% of the articles cited in Wikipedia's health content are reviews, a highly valued source of evidence [81,82]. These studies also assert that a major value of Wikipedia is its curated list of published articles, many of which are cited in Wikipedia within days, but at most months, of publication [79,80].

**Epidemiological studies.** Finally, a little under half of the articles that address Wikipedia's utility within the context of research focused on Wikipedia's potential as a data source in epidemiological research (S5 Appendix, Table 10). Generous, et al (2014) propose that Wikipedia has the ability to respond to the major challenges that currently face disease monitoring–openness, breadth, transferability, and forecasting [83]. There is evidence to support the feasibility of Wikipedia in disease monitoring in relation to seasonal changes in mood [84] and seasonal fluctuations in influenza[85–87]. However, while the work on the development of a plausible means to use Wikipedia's access log data to monitor and forecast disease continues [88], one study actually proposes Google data could be more useful [89] and another study struggles to prove the utility of Wikipedia for this purpose [90], although this is likely due to the fact that the researchers used a sample of 1,633 diseases rather than focus on a single disease as other studies have done.

## Publication trends

In 2018, a study based on a sample of 100,000 articles indexed in CrossRef, a database that indexes all articles that have a DOI, estimated that approximately 28% (95% CI [27.6–28.2%]) of all literature is published using an open access publication model (OA) [91]. Comparatively, a quantitative analysis of the articles and conference proceedings included in this review (Fig 3) indicates that 40.4% (n = 36) are OA. The complement of OA literature within this review is 143% greater than the complement of OA literature indexed within CrossRef, which may indicate that the authors of these studies, like Wikipedia, value openness and the broad dissemination of information. It is encouraging to see evidence of similar values between those investigating Wikipedia and Wikipedia itself.

Another observable pattern is the overall upward trend in the production of health literature about Wikipedia, which is illustrated by a noticeable peak in 2014 (Fig 4), the same year the only two known systematic reviews of Wikipedia scholarship were published [10,11]. The growth of publications in this field is a promising indicator of the growth in the acceptance of Wikipedia within the academic health community.

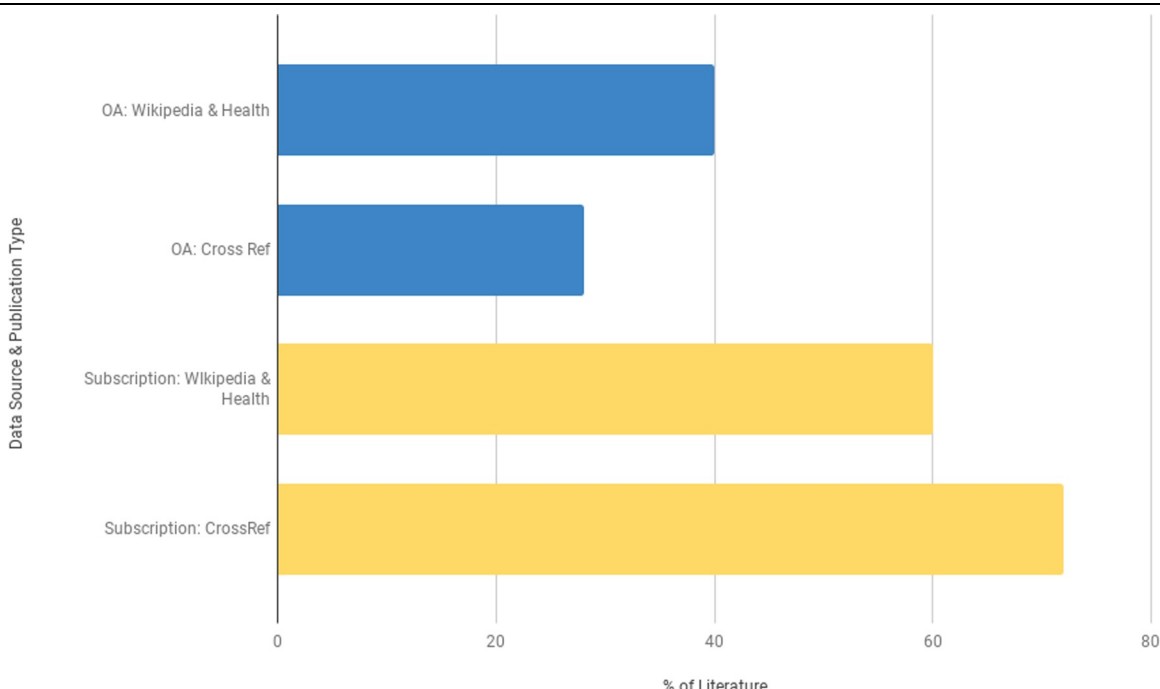

**Fig 3. Distribution of publication models between CrossRef and the health literature on Wikipedia.**

## Discussion

While there are consistent findings with respect to the readability and currency of Wikipedia's health content, the results of assessments of reliability (accuracy, comprehensiveness, currency, and breadth or depth of coverage) are varied at best. These studies vary not only in their results, but also in their methodologies. Studies with dubious methodological approaches assess Wikipedia in comparison to a resource with which significant differences should be expected, such as clinical tools designed for practitioners or experts [27,30,35,36], or websites managed by leading organizations on the topic [41]. More appropriate comparisons are drawn between other open, free tertiary resources [26]. Alternatively, a validated instrument, such as DISCERN, can be used to quantitatively assess Wikipedia articles [33,34,43,92]. Other measures of quality take into consideration the number and quality of references and since there are close to one million references supporting Wikipedia's medical content, this could be a useful approach for the assessment of reliability.

Sample selection methods in studies concerning quality range between using terms derived from course resources [32], semi-structured interviews [43], other tertiary sources [38,39], or national statistics [30] to establish a list of Wikipedia articles to evaluate. One study outright acknowledges that their sample of articles may not be representative of the field in question [93] while another neglects to acknowledge a severe mismatch between the sample studied and the results of the study [94]. In addition, some studies are just too ambitious to possibly be reliable [64,94,95] and of the 47 articles concerned about quality, only four discuss or propose methods for improving or adding to Wikipedia's health and medicine content [62,96–98] in response to their findings. Regardless of methodology or focus, the findings of investigations into Wikipedia's quality are varied with respect to the accuracy, coverage, comprehensiveness of Wikipedia's medical content.

Attempts to generalize findings from a single study to generate an evaluation of the quality of the entire corpus of medical content in Wikipedia are futile at best. It is impossible to

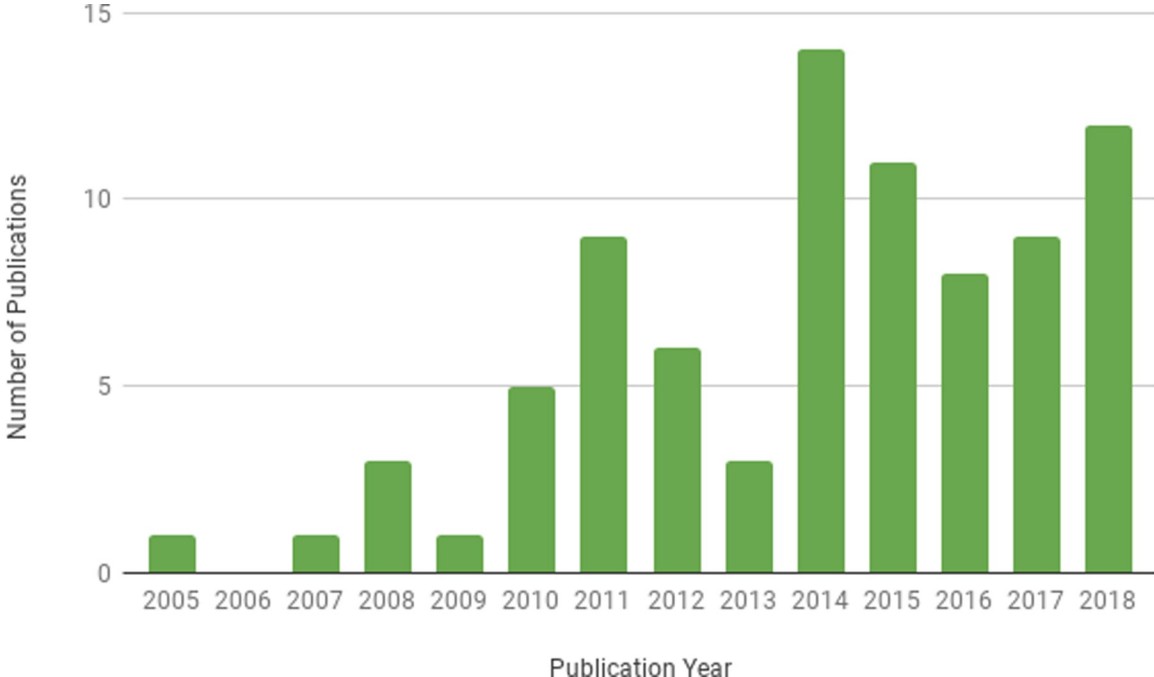

**Fig 4. Distribution of literature by publication year.**

generalize that its medical content, generally, is of either good or poor quality. Wikipedia's articles are individual pieces of a larger whole, creating a mosaic of information where each piece contributes to the summarization of medical knowledge but where some pieces are more complete than others. While it is prudent to describe Wikipedia as a tertiary information resource, it is not equivalent to what is considered a traditional encyclopedia. That is, its content is not clandestinely produced by an elite group of experts and only made available once it is deemed a complete publishable encyclopedia. Wikipedia is fluid. Not only is it incomplete, it is dynamic: evolving, and expanding; thus, rendering the practice of evaluating Wikipedia's content, in any context, extraneous. Any evaluation will remain a snapshot in time that may or may not be applicable by the time the study hits the press. Unless these studies are used to establish a plan to improve Wikipedia's content, it is a challenge to understand their value. Studies that measure the quality of Wikipedia's health information seem to be asking the wrong question. A more appropriate question might be "how can Wikipedia's health information be improved?" Further, recommendations for or against the use of Wikipedia in any context based on quality do not take Wikipedia's utility in education or research under consideration.

Overall the literature tells us that Wikipedia is used in educational and professional environments, it is an engaging and impactful way to teach evidence-based medicine skills or information literacy, that its utility is demonstrated by the knowledge acquisition of students who use it, and that it is difficult to determine whether its quality makes it suitable for student use, despite students being aware of its utility and accessing it frequently. The value of student engagement with Wikipedia also trends in this small body of literature. Students have the time, resources, and capacity to identify gaps in Wikipedia's health content and make the contributions required to improve it. It appears that this approach in health and medical education continues to gather momentum. An interesting contradiction in the literature however, is that some studies find Wikipedia is appropriate for public consumption, while others deem it unsuitable for student consumption This raises the question: what makes Wikipedia suitable

for patient use, but not medical student use? If the Wikipedia community aims to create a Wikipedia that is suitable for *both* medical students *and* patients, what problems may arise?

Finally, there is promise that Wikipedia has potential to benefit the health research community in various contexts (bibliometrics, infodemiology, and epidemiology or disease monitoring), but more needs to be done to develop a better understanding of its utility in this regard. Wikipedia's health content could, potentially, be leveraged to understand the public perception of a specific health topic. An evaluation of online information about genome editing finds that Wikipedia and the top seven web sites retrieved by Google focus primarily on explaining the technicalities of genome editing and do not necessarily focus on other hotly debated conversations around the topic, such as ethics [9]. These findings can be useful in determining what pieces of health information might be missing from public view.

## Limitations

The most significant limitation of this review is its exclusion of literature in which Wikipedia is mentioned in articles that focus on online health information seeking but do not explicitly discuss or address Wikipedia as a major topic. These studies may provide richer insight into the role of Wikipedia in online health information seeking, its prevalence in search results, why it may be selected as a source of health information in online searching, and if, or how, Wikipedia is utilized in conjunction with other online information sources. These are explorations that merit further investigation into existing literature in the future and will be the subject of forthcoming research.

## Conclusion

The literature provides strong evidence to position Wikipedia as a prominent health information resource for the public, patients, students, and practitioners seeking health information online. Wikipedia's health content is accessed frequently, as much as or more than other free online health information sources, such as Medline Plus, and its pages regularly rank highly in search engine results, thus increasing the possibility that a user may select Wikipedia as a health information source. Trends in education where Wikipedia is employed as a tool for education in the classroom, particularly in medical curricula, have emerged but there is no obvious consensus in this domain. Outlier topics such as the utility of Wikipedia in health research or the study of disease outbreaks require more research to produce sufficient evidence to support the value of Wikipedia in research.

Much of the attention on Wikipedia is based on assessments of the quality of its medical content. However, the dynamic nature of Wikipedia and the sheer volume of content that would need to be assessed in order to properly evaluate its medical content holistically often renders these types of evaluations irrelevant. Investigations into accuracy and completeness are particularly difficult to assign value to, but a consistent finding brings to light the low readability of much of Wikipedia's medical content. Rather than rate the quality of Wikipedia within a specific domain and recommend its use or discontinuance of use as an information resource, these findings could instead be utilized as evidence to support advocacy for expert participation in the production of Wikipedia's content. Shafee et al (2017) believe that "Wikipedia is set to retain its position as a key public health information resource" [14] and they issue a call for participation to the medical community, as do their peers [13,25,33], "to ensure that medical content is accurate in the world's most consulted encyclopedia" [14].

While Wikipedia itself is well into its second decade, the academic study of Wikipedia within the context of health is still young, however its use in various contexts is apparent in the existing literature. It can be a patient or consumer health information resource, a resource for

medical or health students, a resource for health and medical professionals, and a data source for researchers.

The literature does not investigate to what capacity Wikipedia is used. While page view data provides evidence that Wikipedia is accessed frequently, this data alone cannot provide insight into what usage looks like. Tom Wilson, a prominent contributor to the field of information behaviour and information needs research is credited for the general notion that "the acquiring of information is not. . . an end in itself. An identification of the use to which [the information resource] is to be put. . . is essential to any form of information seeking" [99]. That is, it is not sufficient to know that Wikipedia is being used for health information. We must also seek to learn to what end it is used. Questions that remain to be answered are: Does the information on Wikipedia influence users' decisions about their own health or health care, or the health care of those within their care, such as children or aging parents? Does Wikipedia's health content, when consumed by patients or the public, supplement, support, or interfere with health care workers' delivery of care to patients? Why is Wikipedia used as a health information resource compared to other free online health information by either patients or health workers? Is it used in conjunction with other online or offline health information resources? What makes a user trust or distrust Wikipedia? What can Wikipedia offer as a health information resource that traditionally accepted online health information resources, such as Medline Plus, cannot? Can the currency of Wikipedia complement the quality of more authoritative health information resources online, and if so, what might this look like?

The next step to contribute to this field that is ripe for deeper investigation will include a comprehensive scoping literature review of online health information seeking behaviour and the role of Wikipedia in the retrieval and use of online health information among patients and health care workers and any differences therein. This review will ask "what is the role of Wikipedia in online health information seeking?" Following the review, an exploratory study into the application of online medical information, with a focus on Wikipedia, will be initiated.

## Supporting information

**S1 Appendix. Search strategy and results.**
(DOCX)

**S2 Appendix. General articles that aim to situate Wikipedia as a health information resource.**
(DOCX)

**S3 Appendix. Studies of Wikipedia content quality categorized by population of concern.**
(DOCX)

**S4 Appendix. Studies of Wikipedia's utility in health education, categorized by education type.**
(DOCX)

**S5 Appendix. Studies of Wikipedia's utility in health-related research, categorized by field.**
(DOCX)

## Author Contributions

**Conceptualization:** Denise A. Smith.

**Data curation:** Denise A. Smith.

**Formal analysis:** Denise A. Smith.

**Funding acquisition:** Denise A. Smith.

**Investigation:** Denise A. Smith.

**Methodology:** Denise A. Smith.

**Project administration:** Denise A. Smith.

**Supervision:** Denise A. Smith.

**Validation:** Denise A. Smith.

**Visualization:** Denise A. Smith.

**Writing – original draft:** Denise A. Smith.

**Writing – review & editing:** Denise A. Smith.

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
