## [Decision Letter · Decision Letter 0]

12 Dec 2019

PONE-D-19-29040

Situating Wikipedia as an online health information resource in various contexts: A scoping review

PLOS ONE

Dear Ms. Smith,

Thank you for submitting your manuscript to PLOS ONE. After careful consideration, we feel that it has merit but does not fully meet PLOS ONE’s publication criteria as it currently stands. Therefore, we invite you to submit a revised version of the manuscript that addresses the points raised during the review process.

Two Reviewers evaluated the manuscript and provided generally favorable opinions. I agree with their evaluation for what regards the content of the contribution, and that a scoping review on this topic is of notable interest. However, I would like to stress the importance of adapting the manuscript to PLoS ONE standards. Specifically, this journal does not consider literature reviews, but it considers scoping reviews given that they are well reported, address a clearly defined research question, have been conducted to a high standard, and are reproducible. Consistently with Reviewer 1 in particular I think the methodological part of this manuscript could be improved.

Revise abstract: background is missing and objective is not clearRevise or improve/specify objectives: while the aim of a scoping review is to “map a field”, this journal requires objectives/research questions to be as specific as possible to make the review reproducibleRedo tables as suggested: Also, some references in the tables include journals, others not – keep consistentMore details and relevant references are needed regarding the exclusion of sources (lines 81-85); it seems reasonable not to include all possible health-related terms, but again make sure enough information is provided to make the review reproducible. The same can be said for the “thematic analysis” (lines 133-139) which lacks methodological details

Finally, Reviewer 2 said his comments are optional, but I suggest you to consider them, especially for what regards correctness about Wikipedia history and activities

We would appreciate receiving your revised manuscript by Jan 26 2020 11:59PM. To enhance the reproducibility of your results, we recommend that if applicable you deposit your laboratory protocols in protocols.io, where a protocol can be assigned its own identifier (DOI) such that it can be cited independently in the future. For instructions see: http://journals.plos.org/plosone/s/submission-guidelines#loc-laboratory-protocols

We look forward to receiving your revised manuscript.

Kind regards,

Stefano Triberti, Ph.D.

Academic Editor

PLOS ONE

2. Please upload a copy of Figure 1, to which you refer in your text on page 6. If the figure is no longer to be included as part of the submission please remove all reference to it within the text. (you appear to have included your figure 1 file as a supporting information file and not a figure)

Reviewers' comments:

Reviewer's Responses to Questions

**Comments to the Author**

1. Is the manuscript technically sound, and do the data support the conclusions?

Reviewer #1: Yes

Reviewer #2: Yes

2. Has the statistical analysis been performed appropriately and rigorously? 

Reviewer #1: N/A

Reviewer #2: Yes

3. Have the authors made all data underlying the findings in their manuscript fully available?

Reviewer #1: Yes

Reviewer #2: Yes

4. Is the manuscript presented in an intelligible fashion and written in standard English?

Reviewer #1: Yes

Reviewer #2: Yes

5. Review Comments to the Author

Reviewer #1: This scoping review studies the peer-reviewed, published health and information science literature about Wikipedia.

The authors’ described exhaustively Wikipedia, its role and its characteristics, in the introduction, which is well written and organized.

Please, correct the citation (6) at line 50 which is reported two times.

The methods are well described; however, I think the authors should explain why they decided to proceed with a scoping review and the methodology to follow for this type of review.

I think that Results are limited; the authors should better explain in which and how many contributions they did find the specific results. The table alone is insufficient, more detailed description in the text should accompany the analysis.

Additionally, the PRISMA figure is not explained in the test: authors should describe the processes that led them to exclude articles from the first exclusion process (they must start from the initial number of articles, 3630, and explain how they arrived at the final number, 89).

Tables should be completely redone. Usually tables in reviews (systematic or scoping) include each individual study (one study per row) and list their characteristics (e.g., population, research design, results relevant for review, etc.) to help the reader identifying relevant information for each study; in the current form, it is really not clear what Authors wanted to explain with such tables. What is the relationship between multiple studies on the same row, for example? Why is it important to divide the studies basing on population especially? Etc.I advise authors to analyze similar reviews from already published literature and to find an alternative for the design of the tables which could be clearer and more informative for the readers.

I think that most of the information relevant for the results were incorrectly placed in the discussion part: information on the retrieved papers should be put in results, while discussion and conclusion should feature critical resume and synthetic reflections about the main take-home messages for the reader. I suggest authors to revise these sections and be clearer in the distinction between a reporting results vs. a discussion section.

Reviewer #2: read the attached PDF

even though I attached a PDF, the form required that I also post some comments here

Excellent draft - if you changed nothing then it is fit for publication. Suggestions below are optional for you to use.

6. PLOS authors have the option to publish the peer review history of their article (what does this mean?). If published, this will include your full peer review and any attached files.

Reviewer #1: No

Reviewer #2: Yes: Lane Rasberry

---

## [Author Response · Author response to Decision Letter 0]

15 Jan 2020

Response to editor:

EDITOR: I would like to stress the importance of adapting the manuscript to PLoS ONE standards. Specifically, this journal does not consider literature reviews, but it considers scoping reviews given that they are well reported, address a clearly defined research question, have been conducted to a high standard, and are reproducible. Consistently with Reviewer 1 in particular I think the methodological part of this manuscript could be improved.

DS: I have revised the manuscript to meet the stylistic and rigor requirements of PLoS ONE to the best that I have understood them based on your feedback and reviewer’s comments. I have made significant improvements to the methods section and added my full search strategy (appendix A). I hope you will find make the study reproducible. 

• EDITOR: Revise abstract: background is missing and objective is not clear

o DS: Added research questions to objective. Added background. 

• EDITOR: Revise or improve/specify objectives: while the aim of a scoping review is to “map a field”, this journal requires objectives/research questions to be as specific as possible to make the review reproducible

o DS: Added research questions to objective in Abstract. RQs are also in line 70 and 71 of introduction.

• EDITOR: Redo tables as suggested: Also, some references in the tables include journals, others not – keep consistent

o DS: Addressed (see response to Reviewer #1)

• EDITOR: More details and relevant references are needed regarding the exclusion of sources (lines 81-85); it seems reasonable not to include all possible health-related terms, but again make sure enough information is provided to make the review reproducible. The same can be said for the “thematic analysis” (lines 133-139) which lacks methodological details

o DS: provided more detail in flow chart (precisely how many articles were excluded for each criterion for exclusion), fleshed out methods section to include rationale for scoping review as the chosen method and a stronger rationale for not including health terms.

o DS: Added more detail to define “thematic analysis” and describe methods of conducting the thematic analysis to improve reproducibility of study.

 Finally, Reviewer 2 said his comments are optional, but I suggest you to consider them, especially for what regards correctness about Wikipedia history and activities

DS: Complete. Responses to individuals comments below (see response to Reviewer #2)

RESPONSE TO REVIEWER 1:

R1: Please, correct the citation (6) at line 50 which is reported two times.

DS: Corrected

R1: The methods are well described; however, I think the authors should explain why they decided to proceed with a scoping review and the methodology to follow for this type of review.

DS: Added search strategy for each database search. Added lines 74-81 to provide context for purpose of scoping review.

R1: I think that Results are limited; the authors should better explain in which and how many contributions they did find the specific results. The table alone is insufficient, more detailed description in the text should accompany the analysis.

DS: I combine the Results and discussion into one section (as in the body formatting guidelines) and was better able to illustrate specifically from where each category emerged. The detailed description in the text that previously sat in the Discussion section now complements the results in a section called Results and Discussion

R1: Additionally, the PRISMA figure is not explained in the test: authors should describe the processes that led them to exclude articles from the first exclusion process (they must start from the initial number of articles, 3630, and explain how they arrived at the final number, 89).

DS: I added a section to the Methods section that describes how I went from my starting number of results to the final 89 articles. NOTE: the numbers changed as I went back to revisit my records and realized I did manual screening of a single database search and this was left out of my originally submitted counts. This has now been accounted for in the text and adequately described.

R1: Tables should be completely redone. Usually tables in reviews (systematic or scoping) include each individual study (one study per row) and list their characteristics (e.g., population, research design, results relevant for review, etc.) to help the reader identifying relevant information for each study; in the current form, it is really not clear what Authors wanted to explain with such tables. What is the relationship between multiple studies on the same row, for example? Why is it important to divide the studies basing on population especially? Etc.I advise authors to analyze similar reviews from already published literature and to find an alternative for the design of the tables which could be clearer and more informative for the readers.

DS: Completely revised tables. I see now how troubling the layout was. Have also renamed title of each of the three tables to be more descriptive about the content therein. Each table is divided into categories and the relevant articles from each category are listed, one study per row. Each study in the table corresponds with a category discussed within the body of the discussion. The purpose of including these tables is to provide readers with a quick reference to consult when they’d like to see all articles that fall within a particular category/theme I identify and discuss in the body of the article. The decision to have some categories labeled by population is because in the body of my discussion I specifically address trends in the literature where some populations are considered more than others. For example, for quality of Wikipedia’s health content, the dominant conversation is around whether the quality makes it suitable for patients, but it is important to provide a list of which articles have evaluated Wikipedia’s quality with patients in mind, or another population, such as medical students or health professionals. 

Since this is a scoping review I chose not to place emphasis on recording study design or other characteristics. The main goal of this review is to get a sense of what conversations are happening in the literature and to draw out any patterns/trends/gaps/or concerns I can find. I made notes when methodology was dubious and report these concerns, when appropriate, in the discussion section. 

I think that most of the information relevant for the results were incorrectly placed in the discussion part: information on the retrieved papers should be put in results, while discussion and conclusion should feature critical resume and synthetic reflections about the main take-home messages for the reader. I suggest authors to revise these sections and be clearer in the distinction between a reporting results vs. a discussion section.

DS: As mentioned above, I moved the discussion section to the results and combined them into one section to provide richer detail in a more streamlined way. The conclusion will include reflections and the main take-home messages.

RESPONSES TO REVIEWER 2:

37 “Imagine a world in which every single person has free access to the sum of all medical 38 knowledge” (1). This is what WikiProject Medicine, a community of editors who develop 39 the health care content in Wikipedia (2), 

Some clarification here - omit the link to Wordpress, your citation #1. This is not maintained and is the same as the #2 citation

https://meta.wikimedia.org/wiki/Wiki_Project_Med

DS: Fixed.

41-42

than any other health information 42 resource online 

"than any other single health information resource"

I realize that we in Wikipedia do not track non-digital media because it is not possible, but Wikipedia is a continually living 20 year global resource. It competes against things like personal doctor visits. 

I think you went to the Google / Facebook / Microsoft event at the WikiConference. I have less understanding of Google, but I think they said that 15% of user queries are health related. I feel increasingly comfortable calling Wikipedia most popular without qualifiers like "online", "new media", etc. 

DS: I removed many of the qualifiers of “online” but not all. I have strategically placed terms like “internet” and “online” in the abstract so that the article can be retrieved in a search for “online OR interent” AND health information. 

However, a paradigm45 shifting study conducted by Nature found that Wikipedia and Encyclopedia 

Britannica 

This was in 2005, we have high school students born after this who use Wikipedia routinely. Since 2005 there has been one human generation of development passed and several technological generations. For example, in 2005 no one thought Facebook would become popular. . There has been no big challenge of Wikipedia since then. There have been various small challenges. 

You know the literature. My point in this is just that the Nature study is old and the bigger news is that no one ever challenged it. 

DS: Yes, I included this study so that other readers are aware I know it exists. A common exercise in academia, I’m afraid. You’re right to point out that the bigger news is that the finding has not been challenged and so I have added a sentence to that effect. Line 47-48

59 Today Wikipedia is the 7th most frequently accessed web site in the United States and 60 Canada(7). 

Since 2005 Wikipedia has been an Alexa top 10 website. There is no source to cite for this. Evidence becomes stronger after 2007 and stronger again after 2012, but no one has published this simple reporting. If you feel bold enough you could say "top ten website since at least 2014" and cite Heilman 2015 https://www.ncbi.nlm.nih.gov/pmc/articles/PMC4376174/

the strength of the claim in the source is weak but the statement is accurate 

DS: I cited the Alex web site rankings: https://www.alexa.com/topsites/countries/US & https://www.alexa.com/topsites/countries/CA. There are new stats, so I updated the manuscript with the latest rankings for US and Canada

Wikicite has 90 papers, you found 89, and I wonder what the difference is. 

DS: It’s possible new articles have been published since I ran my searches 7 months ago. In fact I know there have been at least one or two new articles about editing Wikipedia or evaluation quality of Wikipedia, I just coudln’t add them to this review retroactively and if I did, they wouldn’t change my findings anyhow. 

98 treatment of Wikipedia's health content was insufficient to merit inclusion. These articles 6 99 will be reviewed separately in an additional study of the literature that specifically explores 100 online health information seeking behaviour and Wikipedia’s role within it. 

I get anxious seeing promises like this made because it can be hard to identify the later paper if it exists. I sort of wish this promise could be omitted. 

DS: You’re right. It’s a bold statement. I have removed. But for the record – I did end up doing this work. If it gets published I’ll be sure to cite this review so readers can draw the connections on their own.

110 In 2018, a study based on a sample of 100,000 articles indexed in CrossRef, a database that 111 indexes al 

This bit about OA is interesting but this is the start of the results section, and I was hoping for a strong start about Wikipedia. The OA information could come later as it is more incidental. 

DS: Yes, it’s jarring that I don’t launch into results about Wikipedia straight away. I have moved the OA content and have moved it to the end of the Results and discussion section. 

117 openness and the broad dissemination of information. It is encouraging to see evidence of 118 similar values between those investigating Wikipedia and the organization itself. 

Can you clarify what "the organization" is? Perhaps rephrase? 

DS: Yes – changed “the organization” to “Wikipedia” since it is not an organization, but a project of the Wikimedia Foundation

124 literature about Wikipedia is illustrated by a noticeable peak in 2014 (Fig 3), the same year 125 the only two known systematic reviews of Wikipedia scholarship were published. 

Wait - can you cite the reviews to which you are referring? 

DS: Done. 

125 the only two known systematic reviews of Wikipedia scholarship were published. In 2015, 126 Nature reported its findings that suggested Wikipedia was more or less equal in accuracy 127 compared to Encyclopedia Britannica 

Seems like an error! Not 2015, 2005! 

https://www.nature.com/articles/438900a

I am not aware of a 2015 paper! 

DS: Yes this was a typo I became aware of after submitting the manuscript. Thanks for the reminder. I may have forgotten to fix it otherwise.

123 -129

perhaps, but is this a confusion about the date of the nature paper? If so then revise or strike this. 

DS: fixed… a result of my typo and working on this too late at night!

134 1. Articles with a general focus on situating Wikipedia as a health information resource 135 with no specific context as a major focus (e.g for consumers, professionals, or 136 students); 

It seems extraneous to qualify this when this is the only category of this sort. Why not just say "Articles with a general focus on situating Wikipedia as a health information resource", then later explain this additional detail that you sought no specific audience targeting? 

DS: This is reasonable to me. I’ve adjusted. I do explain this later so you’re right to point out the extraneity of it

156 dominate the conversation. 53% (n=47) of the articles included in this study measure 

Is "measure" the correct term? I would have expected "report" or "make claims upon", but I think most studies do not do a measurement in a quantitative way. 

DS: Changed to “assess”

160 currency, comprehensiveness, completeness of coverage, number of references, and 161 currency. 

currency twice

also I know that Wikipedia emphasizes number of references, but I am not aware of other sources strongly valuing citations in medical information like Wikipedia does. 

DS: Removed duplicate word. Some studies in this review evaluate Wikipedia’s quality by assessing the number of references its pages on a specific topic have. This is why it is included in the list of characteristics that may be used to evaluate the quality of Wikipedia.

181 resource demonstrate this by references its frequency of use and some advocate for the 182 medical community to contribute to Wikipedia’s medical content. E 

something unclear here - "references" to "referencing", perhaps 

182 medical community to contribute to Wikipedia’s medical content. Evidence has been 183 collected from the Pew Research Centre 

This confuses me a bit. Active voice could help - 

DS: Yes- revised the above two statements with more clear, direct language. 

218 same topics (17). This is study is seminal positioning Wikipedia’s prominence in the 219 existing complement of online health information resources. 

rephrase

This study positions Wikipedia as prominent among the existing... 

 DS: done. 

227 gastroenterology (18), nephrology (19), and some cancers (20–22). These assessments 228 assess 

"assessments assess" to "studies assess" or similar 

DS: “assessments” changed to “studies” 

229 findings in relation to the public consumer or patient. Of those that include results – some 230 conference proceedings do not – they generally agree that Wikipedia is suitable for patients 231 (18,19,22) and a 2010 study find 

Cut this into multiple sentences. The hyphen conjuctions etc is too much. 

DS: done.

232 the preferred one (21). Unfortunately, these assessments of suitability for patients are no 233 longer likely to be relevant with publication dates ranging between 2008 and 2013. 

The situation is mixed. Many articles require updating and some do not. 

DS: Yes – some articles require updating and some do not. Howeever, these particular assessments are mainly concerned with whether or not the content is suitable for patients. The do not necessarily provide evidence to suggest that an outcome of their findings will be to update the content. Some call on the medical community to contribute, others don’t even mention the possibility of making changes to the articles they assess. Also – the studies of suitability are also unlikely to be relevant today because they were published years ago. We can’t evaluate Wikipedia articles based on studies 6+ years old. 

236 (2017) investigation into who edits Wikipedia’s medicines content "medicines" to medical, as this not about drugs specifically 

DS: This is the language used in the article and it is referring to pharmaceutical medications. Revised language to be more clear: “Wikipedia’s content on medicines”

248 Two distinctive studies describe leveraging Wikipedia’s content in order to improve 249 patient care and this is perhaps the most exciting finding of this review. 

I do not follow why you think these are the most exciting. Can you explain more? 

DS: Added: “due to their real world application to medical practice” I found these studies exciting because they provided examples of broad uses of Wikipedia for patient care beyond physicians using Wikipedia articles to jog their memory.

251 now integrated with Wikipedia to pull radiology images from the WikiMedia Commons into 

It is stylized "Wikimedia Commons" 

DS: Revised

252 their database (46). Another study describes how it used Wikipedia to mine consumer 253 health vocabulary by finding synonyms for medical terminology 

Excavating the mother lode 

https://www.sciencedirect.com/science/article/pii/S0306457316303004

This talks about medicine and the people behind this are wiki medicine people. Not sure if you detected this one as it does not have Wikipedia or medicine in the title. The paper was bold enough to identify Wikipedia as the mother lode but regrettably did not say Wikipedia in the title. They really are sitting on a big idea.

That was out of scope for you but confirms the application. 

DS: I did see that that has Wikipedia in the title and this article. It was retrieved in my Web of Science search. It was excluded in the title/abstract round because it did not mention health or medicine. These categories threw me off: “information retrieval, natural language processing, and ontology building”and made it sound like a paper focused on computer science applications. This review cited a study from Friedlen and McDonald (2010) that I had found but determined irrelevant due to its IT/CompSci focus. You’re right that the review article out of scope but is a great confirmation of application of this practice.

262 the findings cannot be generalized (48). Interestingly this study also finds that of the 263 participants, 65% did not know how to revise Wikipedia when they encountered an error. 

prefer "did not know that Wikipedia invites them to edit text to revise errors which they encounter" 

DS: Not changed. This is a direct quote from the study: “Of those, 861 (65%) students did not know how to revise articles and 199 (15%) let the false information unaltered, despite knowing how to correct articles” Changing the sentence according to your suggestion above would not be an accurate representation of the findings in the cited paper. The study did not report on whether the participants knew that they could edit, just that they did or did not know how to do it.

277 studies, and the findings vary from one study to another. Some conclude Wikipedia is 278 suitable for medical students(53,57), but most conclude it is not(53–56,58,59). 

yeah, I wish more studies would compare Wikipedia's suitability to any other specific thing instead of comparing it to the ideal of perfection. 

DS: Right – the reality is there is no “gold standard” and Wikipedia cannot be compared to perfection or the “ideal information resource” which inherently does not exist. All types of information resources have their own unique strengths and flaws.

That project, Wikipedia Zero, is sunsetted. Regardless of publication and evaluation, we in the Wikipedia editing and outreach community never became aware of any cases of use which resulted in anyone contacting anyone in our social network. It was kind of an expensive outsider project which never had Wikipedia community collaboration in an apparent way. 

The community effort, named as a flagship Wikimedia Foundaiton accomplishment in their 2018 annual report, is Internet in a Box. I and lots of WikiProject Medicine people contribute to this. https://meta.wikimedia.org/wiki/Internet-in-a-Box

I wish you could omit this text just because that Wikipedia Zero project is over, it never matured, it had more presence in publication than in Wikipedia community engagement, and it was kind of a project on paper which had thin ties to any reality. I do not want to criticize it because it could be revived, good people contributed to it, and is still a good idea but we do not have much to show for this. 

DS: I am aware of the Internet-in-a-Box and its applications. The reference to Wikipedia Zero here is entirely relative to the specific study I’m reporting on, where the value of Wikipedia to this community of Botswanan medical workers was dependent on the fact that they could access it without internet. That despite awareness of Wikipedia’s flaws, they still saw it as valuable simply because it was consistently available. So – rather than omit this section, I have added some commentary, and citations, that draw to the light the fact that Wikipedia Zero does not exist any longer. 

314 that while Wikipedia does well to remain current (27,42,44), the readability of its medical 315 content is too low to accommodate the very people it stands to benefit most. The lowest 316 reading level reported is around ninth grade (26,31,41) 

clarity here please 

you use the term "low" in two different senses. One sense is "low readability", which I think means "high complexity", and the other sense is "low reading level", which means "low complexity". 

perhaps 

that while Wikipedia does well to remain current (27,42,44), its medical content uses technical terms which result in readability which is too low accommodate the very people it stands to benefit most. The most accessible reading level reported is around ninth grade (26,31,41) 

DS: Thanks. Changed this to: “ while Wikipedia does well to remain current (27,42,44), its medical content uses technical terms that result in readability levels too low to accommodate the very people it stands to benefit most. The readability of the easiest articles has been reported to be around ninth grade (26,31,41)”

325 methodological approaches assess Wikipedia in comparison to a resource with which 326 significant differences should be expected, such as subscription-based reference tools 327 designed for practitioners or experts (28,31,36,37), or websites managed by leading 328 organizations on the topic (42) 

I think it is fair to compare Wikipedia with leading resources, regardless of whether they claim to have good subscription funding or target expects. The dubious methodologies include things like setting comparisons where Wikipedia does not try to be competitive, like taking points off Wikipedia for not including dose information or professional practice information. Wikipedia competes as reference information and should be compared on that basis. 

DS: Yes- this is what I am getting at in this sentence. That comparisons should not be drawn between Wikipedia and a resource that is designed for something entirely different than Wikipedia. Since Wikipedia is meant to summarize medical information, we cannot expect it to be as robust as a textbook on how to perform vascular operating procedures, or as you say, dosage information. I have revised the sentence to be more clear about my intent.

328 More appropriate comparisons are drawn between other 329 open, free tertiary resources (27) or, alternatively, a validated instrument, such as 330 DISCERN, is used to quantitatively assess Wikipedia articles (34,35,43,63) 

two kinds of comparisons - reference information quality is one issue, and accessibility is another issue. For quality I want Wikipedia compared against the best competing resource for any topic. Along with that comparison, there can also be a comparison of accessibility. 

Wikipedia could then also be compared against the best available free resource, but I see little value in that because I expect that nothing would come close to Wikipedia's quality. Also any other available free resource is almost certainly negligible traffic and therefore inaccessible. 

DS: I agree, but as this is a review I can only report on what the literature has reported and at this point, many of the published studies that compare Wikipedia to another resource seem to missing the point – that Wikipedia is meant to be an encyclopedia. It is a tertiary source. Therefore, it is inappropriate to draw comparisons between the “quality” of a an encyclopedia compared to a textbook or clinical tool, which are an entirely different type of information resource. For example, I wouldn’t go to Dynamed or UpToDate to get an overview of a medical condition for the same reasons I wouldn’t seek out dosage guidelines on Wikipedia. Each type of research has its own role to play in the landscape of medical information and the comparisons published to date don’t seem to understand that nuance. 

347 Any attempt to generalize findings in order to generate a holistic understanding of the 348 quality of the entirety Wikipedia’s medical content is an insurmountable task. 

specific the methodology which you imagine 

I admit that we do not have a holistic understanding but also no one has tried. I think if we had modest funding to do an evaluation then we could do well to get a good general understanding. There are several possible strategies to attempt and realistically there has hardly been any attempt. 

I am much more hopeful about this. 

DS: What I meant here is that some studies included in this review evaluated the quality of a sub set of Wikipedia articles on a given topic (e.g. autoimmune disorders) and then generalized those findings to make claims about the quality of Wikipedia’s medical content more broadly or at least more broadly than the articles included in the study. Attempts to generalize findings to generate a holistic view of the quality of Wikipedia’s medical content is futile. The articles are changing all the time and the quality of articles is not standard across Wikipedia, Some are poor, some are remarkably well done. So it’s impossible to say whether or not Wikipedia, generally, is “good” or not. I hope the revision makes this more clear:

“Attempts to generalize findings from a single study to generate an evaluation of the quality of the entire corpus of medical content in Wikipedia content are futile at best. It is impossible to generalize that its medical content, generally, is of either good or poor quality. Wikipedia’s articles are individual pieces of a larger whole, creating a mosaic of information where each piece contributes to the summarization of medical knowledge but where some pieces are more complete than others.”

351 more complete than others. Wikipedia is also fluid. While it is prudent to describe 352 Wikipedia as a tertiary information resource, it is not equivalent to a published 353 encyclopedia. Not only is it incomplete, it is also dynamic: evolving, and expanding; thus 

It has been 20 years. Wikipedia is the golden standard for what defines a published encyclopedia. Saying that it is "not equivalent to a published encyclopedia" is odd in the context that Wikipedia is known as an encyclopedia and gets more human hours of consultation than all other encyclopedias put together. At some point it becomes necessary to state what one expects of an encyclopedia and what Wikipedia is. 

DS: Thanks for pointing out the need for me to clarify this. By “traditional encyclopedia” what I really meant is that while it IS an encyclopedia, it is not traditional in the sense that is not clandestinely laboured over for years by an elite group of hand-picked experts and then published when it is deemed to be complete. Have made the change to this: 

“While it is prudent to describe Wikipedia as a tertiary information resource, it is not equivalent to what is considered to be a traditional encyclopedia. That is, its content is not clandestinely produced by an elite group of experts and only made available once it is deemed as complete published encyclopedia. Wikipedia is fluid. Not only is it incomplete, it is dynamic: evolving, and expanding; thus, rendering the practice of evaluating Wikipedia’s content, in any context, extraneous”

Also remember that yes, Wikipedia is 20 years old, but its prevalence in the academic sector is still young. Anecdotally, I still have yet to meet a university student who WAS NOT told by their high school teachers to avoid Wikipedia because it is not reliable. So it’s been around, yes. We (Wikipedians) know it’s the new model of encyclopedic knowledge. But academia is still catching up.

357 Wikipedia’s content, it is a challenge to understand their value. Studies that measure the 358 quality of Wikipedia’s health information seem to be asking the wrong question. Further, 359 recommendations for or against the use of Wikipedia in any context do not take into 360 consideration Wikipedia’s utility. 

Be direct, what is the wrong question, and what is the right question 

DS: “Studies that measure the quality of Wikipedia’s health information seem to be asking the wrong question. A more appropriate question would be “how can Wikipedia’s health information be improved?” – Added.

368 The first known for-credit course at a medical school for editing Wikipedia was offered by 369 University of California, San Francisco in 2013 (50), 

"The first documented record of a for credit course at a medical school" 

I led a Wikipedia editing series at the University of Washington medical school in 2011. When I did that there were a few others doing similar things. When that 2013 program began I trained Amin Azzam based on the culture of practice for medical editing in schools which already existed for this. 

I guess that I am realizing that I need to write up some history because many people at many institutions contributed greatly, and I want the origin story to be Wikipedia-style normal community collaboration and not one person's new idea. 

DS: Changed langage to “documented.” You’re right. There’s also been a long history of editing in undergraduate education, but not necessarily as a full course. It would be neat to create a wiki timeline of Wikipedia editing in medical education that all educators who have used Wikipedia could contribute to.

369 University of California, San Francisco in 2013 (50), but the first reported use of Wikipedia 370 in a course, as a single assignment, is from 2011 and it required students to edit 371 neuroscience stub articles (71) 

You might mean "medical course". There are probably earlier records of medical school course editing on Wikipedia, but not in academic publishing. Likewise there are earlier records of class editing. United States Public Policy in 2010 was the first big classroom initiative. https://en.wikipedia.org/wiki/Wikipedia:WikiProject_United_States_Public_Policy

even this followed earlier classroom experiments. 

DS: changed course to “medical course” 

372 education courses has crossed a number of health disciplines. Pharmacy students have 373 been asked to write content in Wikipedia as an alternative to composing pharmaceutical 374 drug monographs (70) and a gerontology instructor identified gaps in Wikipedia’s content 375 on aging and assigned students the task of using their term papers to edit it (72). 

Here is my own study about classroom use. If you are looking for examples of classroom engagement then this program is in its 5th year and of course this paper is Wikipedia-values oriented. 

Improving the Quality of Consumer Health Information on Wikipedia: Case Series 

https://www.jmir.org/2019/3/e12450/

DS: Thanks – this article was selected for inclusion in the review. I cited it after this sentence 318-319: “only four discuss or propose methods for improving or adding to Wikipedia’s health and medicine content(62,68–70)”

394 to UpToDate and a digital textbook 

check your consistent stylization UpToDate 

DS: used Find and Replace function to fix this. Thanks. 

441 There is a notable trend in the assertion that one major value of 442 Wikipedia is its curated list of published articles, m 

not sure what this means, clari fy please 

DS: Changed to “These studies also assert that a major value of Wikipedia is its curated list of published articles, many of which are cited in Wikipedia within days, but at most months, of publication(81,82).” 

Hope this helps

458 context of research focused on Wikipedia’s potential role as a data source in 459 epidemiological research. Generous, e 

you mentioned the Ebola study above. Maybe that is better placed here? The context of that was that the WHO did not have expertise to translate Ebola text into languages of Africa, but Wikipedia did have community of translators, and a platform to accept this language content, and the means of distribution for this information. If you want an epidemiological example then that ebola incident is the one the media picked up. 

DS: I prefer to leave the reference to ebola where it is. It is actually a citation from Heilman’s “evolution” article, and not a study on ebola and Wikipedia. The context of the Ebola example was to demonstrate the impact that Wikipedia Zero had during the Ebola outbreak. So, I mentioned Ebola as an interesting example of Wikipedia’s global impact. 

470 As promising as the research into the potential utility of Wikipedia in epidemiology may be, 471 one major limitation of using Wikipedia for outbreak prediction is often touted as one of 472 Wikipedia’s strengths: a 

This is a misunderstanding. Google will not talk to university researchers. Google is inhuman and like aliens far away. Wikipedia is a human project and of course we have the data to emulate Zeiger's Google Flu Trends project. Typical researchers can also ask for some private IP address data. Wikipedia does not share this data lightly, but whereas Google almost certainly would not share data with less than a ~300 million investment in this, the buy in for access to Wikipedia's data is ~1 million and commitment to act like a human while having conversations in public with the wiki community. Of course we want to do this project, and the barrier is not on the Wikipedia side of this. 

DS: Thanks for pointing this out. I was under the incorrect impression that Wikipedia does not collect data about the locations of its users, period. And that data related to pageview statistics is completely anonymized. I appreciate that you’ve pointed out that data can be made available for research at the discretion of Wikipedia (for a price). This is something I’d like to learn more about. I have removed the following:

“As promising as the research into the potential utility of Wikipedia in epidemiology may be, one major limitation of using Wikipedia for outbreak prediction is often touted as one of Wikipedia’s strengths: all global searches of Wikipedia are aggregated so searches that originate in a specific geographical location cannot be parsed from the data. Unless we can isolate pageviews geographically it will be difficult to determine a geographic location for disease outbreak.” 

And I have replaced it with:

“Overall, there is promise in Wikipedia for epidemiological data, but more needs to be done to develop a better understanding of its utility in this field.”

489 than other free online health information sources, such as Medline Plus, and its pages 490 regularly rank highly in Google search results, 

you could say Bing and DuckDuckGo, or you could say search engines. Google is not quite universal. Russia and Korea use other search engines, and lots of other countries do too. China is moving onto the world stage and probably will compete against Google in the developing world with their search engines eventually. If you generalize this paper has some more life and citability in the future. 

DS: Good point. Changed.

512 within the context of health is still in the early stages of its development. Several questions 513 have not yet been asked within the literature that are instrumental in understanding 514 Wikipedia’s role as a health information resource and various contexts: as a patient or 515 consumer health information resource, as a resource for medical or health students, as a 516 resource for professional in health and medicine, and as a data resource for researchers. 

can you rephrase? this is long and blocky and I cannot readily see what questions you want asked and answered. 

DS: Completely rephrased this to: 

“While Wikipedia itself is well into its second decade, the academic study of Wikipedia within the context of health is still in young. However, its role in various contexts is apparent in the existing literature. It can be a patient or consumer health information resource, a resource for medical or health students, a resource for health and medical professionals, and a data source for researchers”

520 explaining the technicalities of genome editing and do not necessarily focus on other hotly 521 debated conversations around the topic, such as ethics (9). 

Obviously the issue is that the genetics people import structured data at scale, which is relatively easy, whereas for ethics we require prose which is more difficult. 

DS: Yes. Since I am reviewing the literature I don’t feel comfortable making that kind of claim. I don’t know enough about the project to do that.

525 data provides evidence that Wikipedia is accessed frequently, this data is insufficient to 526 understand whether its content is actually used by the consumer. 

I know why you say this because everyone says this but it is totally bunk. No one holds other communication channels to this standard. Everyone assumes that doctor's office pamphlets and billboards and online patient guides and apps andWe have no reasonable evidence of how consumers use other sources. Suddenly when Wikipedia appears among other communication resources somehow people question whether users get impact out of Wikipedia while taking for granted that they get impact from other communication channels. It is not reasonable and it indicates a bias against Wikipedia. 

You are not citing sources on this so if this is your own musing, I would prefer that this be phrased like this. 

There is evidence that Wikipedia is more popular than other channels, but some critics dismiss this by saying that the value of measurable user attention to Wikipedia is less than the value of user attention to other sources. It would be useful to have a system to assign impact value consistently to Wikipedia and other outreach channels so that their usefulness could be compared. 

DS: You are correct that this is my own musing and I will rephrase so that it is more palatable. This is basically what I will be doing my dissertation research on because in a field as evidence-based as health and medicine, we don’t have qualitative evidence to describe how Wikipedia is used, where it fits in the learning process of patients, etc. I have rephrased to: “While page view data provides evidence that Wikipedia is accessed frequently, this data alone cannot provide insight into what usage looks like.”

Figure 2 

Can you label this more explicity? Perhaps you do this in the text of the article. Just looking at the image I became lost understanding the vertical access labels. 

DS: I have revised all labels in the article to be more explicit.

Figure 4

You might have already described this but I missed the place. Was the classification obvious or did you use a non-obvious sorting system? 

DS: I have revised the methodology section, which now indicates that I categorized articles inductively

---

## [Editor Report · Decision Letter 1]

21 Jan 2020

PONE-D-19-29040R1

Situating Wikipedia as a health information resource in various contexts: A scoping review

PLOS ONE

Dear Ms. Smith,

Thank you for submitting your manuscript to PLOS ONE. After careful consideration, we feel that it has merit but does not fully meet PLOS ONE’s publication criteria as it currently stands. Therefore, we invite you to submit a revised version of the manuscript that addresses the points raised during the review process.

The Author properly responded to Reviewers' concerns. Minor modifications are still needed: 

- abstract has been revised well for what regards content and structure, but now it exceeds the 300 words lenght allowed: https://journals.plos.org/plosone/s/submission-guidelines

- The Author declared in the cover letter being awarded a grant from the Wikimedia Foundation whose funding would cover this publication if accepted. Financial Disclosure section and Competing Interests section should be updated accordingly (in competing interests, I believe Author could specify having obtained the grant -after- completing and submitting this work)

We would appreciate receiving your revised manuscript by Mar 06 2020 11:59PM. To enhance the reproducibility of your results, we recommend that if applicable you deposit your laboratory protocols in protocols.io, where a protocol can be assigned its own identifier (DOI) such that it can be cited independently in the future. For instructions see: http://journals.plos.org/plosone/s/submission-guidelines#loc-laboratory-protocols

We look forward to receiving your revised manuscript.

Kind regards,

Stefano Triberti, Ph.D.

Academic Editor

PLOS ONE

---

## [Author Response · Author response to Decision Letter 1]

22 Jan 2020

EDITOR: The Author properly responded to Reviewers' concerns. Minor modifications are still needed: 

- abstract has been revised well for what regards content and structure, but now it exceeds the 300 words lenght allowed: https://journals.plos.org/plosone/s/submission-guidelines

DS: My mistake. I thought I had double checked this against the word limit. I have revised and reduced abstract to 295 words.

- The Author declared in the cover letter being awarded a grant from the Wikimedia Foundation whose funding would cover this publication if accepted. Financial Disclosure section and Competing Interests section should be updated accordingly (in competing interests, I believe Author could specify having obtained the grant -after- completing and submitting this work)

DS: During the resubmission process, I was not granted access to update the Financial Disclosure section. I emailed PLoS ONE and was advised to provide the update to my financial disclosure statement in my cover letter ONLY, which I did. Again, I am unable to access the Financial Disclosure section through the resubmission process. Instead, I have drafted the disclosure I would have included in this section, and have included it in a new cover letter.

I have added the statement suggested by the Editor (above) to the Competing Interests section of the paper.

---

## [Editor Report · Decision Letter 2]

24 Jan 2020

Situating Wikipedia as a health information resource in various contexts: A scoping review

PONE-D-19-29040R2

Dear Dr. Smith,

We are pleased to inform you that your manuscript has been judged scientifically suitable for publication and will be formally accepted for publication once it complies with all outstanding technical requirements.

With kind regards,

Stefano Triberti, Ph.D.

Academic Editor

PLOS ONE
---

## [Editor Report · Acceptance letter]

30 Jan 2020

PONE-D-19-29040R2 

Situating Wikipedia as a health information resource in various contexts: A scoping review 

Dear Dr. Smith:

I am pleased to inform you that your manuscript has been deemed suitable for publication in PLOS ONE. Congratulations! Your manuscript is now with our production department. 

With kind regards,

on behalf of

Dr. Stefano Triberti 

Academic Editor

PLOS ONE